# An Investigation on the Feasibility of Fabricating Composites Using Outdated Waste Carbon Fibers and Easily Disposable Polyolefin Resins

**DOI:** 10.3390/polym13172938

**Published:** 2021-08-31

**Authors:** Dong-Jun Kwon, Kang Rae Cho, Hyoung-Seock Seo

**Affiliations:** 1Research Institute for Green Energy Convergence Technology, Gyeongsang National University, Jinju 52828, Korea; rorrir@empas.com; 2Department of Chemical and Biological Engineering, College of Engineering, Sookmyung Women’s University, Seoul 04310, Korea; 3School of Naval Architecture and Ocean Engineering, University of Ulsan, Ulsan 44610, Korea

**Keywords:** olefin resins, outdated-waste-carbon-fiber-reinforced composites, mechanical properties, recycling, environmental aspects

## Abstract

Outdated-waste-carbon-fiber-reinforced olefin composites (oCFOCs) were fabricated with easily disposable polyolefin resins, polypropylene (PP), high-density polyethylene (HDPE), and low-density polyethylene (LDPE), by compressive molding using a hot press. The flexural and impact strengths of the oCFOCs from each respective resin type and oCF content, ranging from 35 to 70 wt.%, were increased by the aging treatment (120 °C and 95% humidity under a pressure of 0.8 MPa) until an aging time of three days, due to improved resin impregnation. For the oCFOC with PP, the hydrogen bond between PP and developed C-O groups due to the aging treatment and the existing silane layer of oCF is considered to assist cohesion between the resin and oCF. In particular, PP and 45 wt.% oCF content were the most effective conditions for improving the oCFOCs’ mechanical properties, in addition to endowing the oCFOCs with good moldability and dimensional stability. Our results demonstrate that durable recycled composites can be manufactured using oCF and PP.

## 1. Introduction

Waste generation from everyday life and industry increases continuously as industry develops, arousing great attention for recycling wastes for the environment [1,2]. Among wastes such as paper, metal, and plastic materials, the problem with plastic waste is intensified with the increasing demand for delivery and take-out services as the COVID-19 virus spreads globally. The production of disposable goods and subsequent plastic waste in 2020 was increased by up to 10–20% and 62%, respectively, compared to the figures from 2019 [3,4].

Due to the seriousness of the impact of plastic waste on our environment, significant efforts have already been made to recycle it. For example, Dave Hakkens, an industrial designer from the Netherlands, has been developing the so-called ‘Precious plastic project’, where small-scale plastic workshops set up by individual consumers create new products using recycled plastic [5]. In this context, much attention has also been given to recycling composite waste, especially that deriving from the thermoplastic-based fiber-reinforced composite which generates much waste due to its increasing demand and application [6]; in particular, as the use of carbon fibers (CF) as reinforcing materials expands, a large amount of wastes, including CF, pre-impregnated carbon fibers (prepreg), and resins, is generated [7,8]. Moreover, CF and prepreg, which have a limited warranty period, can become outdated waste before use if they are not properly managed.

Although outdated waste carbon fiber (oCF) is generally not suitable for manufacturing structural parts, sectors, or high-performance components due to the poor interfacial adhesion with the resin matrix caused by the self-reaction of sizing on the surface of oCF [9,10], it has superior mechanical properties compared to other low-priced reinforcing fibers such as glass fiber and low-quality CF types [11,12,13]. Because of this merit, it can be used to manufacture low-cost composite products after proper recycling processes via physical or chemical methods [14,15], drawing attention to its reuse as recycled oCF. The oCF obtained through these recycling processes is commercially named recycled CF (rCF) and sold to the world [16]. rCF is currently used as reinforcing short fibers, utilizing thermoplastic resins of the matrix material to allow for the manufacturing of low-cost composite parts by high-speed molding [17,18,19].

For the thermoplastic resins used as base materials for the composites, thermoplastic olefin resins show good performance in manufacturing large-area composite structures requiring tensile strengths over 180–240 MPa by high-speed molding [20,21]. Among the thermoplastic olefin resins, polypropylene (PP) is the most basic resin, and polyethylene (PE)-based resins such as high-density polyethylene (HDPE) and low-density polyethylene (LDPE) are also available [22,23,24]. Since these resins are inexpensive (subsequently easily disposable) and manifest substantially improved mechanical properties when combined with CF, together with good performance in the high-speed molding, it is worth manufacturing low-cost composites by using these thermoplastic olefin resins, outdated waste CF (oCF), and/or recycled CF (rCF) in the light of recycling and environmental aspects [25,26].

In this study, first, as a way of evaluating the mechanical properties of oCF, i.e., the reinforcing material, the tensile strengths of single-fiber neat CF and oCF were compared. Then, the feasibility of fabricating outdated-waste-CF-reinforced olefin composites (oCFOCs) with PP file covers, HDPE vinyl bags, or LDPE zipper vinyl bags was examined by testing their molding characteristics together with mechanical properties and aging stabilities under specific aging conditions (120 °C, 95% humidity, and a pressure of 0.8 MPa). The results show that the oCFOCs with PP or LDPE could be fabricated with stable moldability while the oCFOCs with HDPE could not. The flexural and impact strengths of the composites fabricated from each respective resin type and oCF content, ranging from 35 to 70 wt.%, were increased under the aging treatment up to the aging time of three days, mainly due to improved resin impregnation; these results show that the mechanical properties of the oCFOCs could be increased by the aging treatment with suitable hot and humid conditions. For oCFOCs with PP, hydrogen bonds that formed between the resin and existing silane layer of oCF were also considered to contribute to the enhanced mechanical properties in addition to the improved impregnation of the resin into the oCF tow. Among oCFOCs fabricated from each respective resin type and various oCF contents, oCFOCs with PP and 45 wt.% oCF content with three aging days exhibited the highest flexural and impact strength; even the interfacial shear strength was about 70% higher than that of the as-prepared composite with neat PP. Thus, the results show that durable oCFOCs with good mechanical properties could be manufactured by choosing a proper matrix (such as PP for this study) together with optimal oCF contents under suitable aging conditions. We propose that our results will serve as important preliminary data to demonstrate and support the potential of fabricating durable oCF composites, as a beneficial method of oCF recycling.

## 2. Materials and Methods

### 2.1. Materials

Three types of commercial olefin resin, which are easily disposable in daily life, were used as matrix materials to fabricate oCFOCs: (1) PP file covers (A4 L holder, MEDIA Co., LTD, Anyang, Korea), (2) HDPE vinyl bags (Weifang Kang Plastic Products Co., Shanghai China) and (3) LDPE zipper vinyl bags (pew28-38, Enif Co., Ltd., Pyeongtaek, Korea). These products are produced and consumed in significant quantities around the world. The respective thickness of all three olefin resins was 40 μm. The oCF (A-42, DOWAKSA Co., LTD., Tuscon, AZ, USA) existed in the woven state and included many specks of dust on its surface (see red circles in Figure 1a).

### 2.2. Methodologies

#### 2.2.1. Fabrication of oCFOCs

The fabrication process of oCFOCs is schematically expressed in Figure 1b. The size of individual specimens applied to the molding process of the oCFOC fabrication was 200 mm × 200 mm × 3.5 mm. The olefin films and oCF fabrics were stacked as shown in Figure 1b. The specimens were prepared with different weight ratios of oCF (35, 40, 45, 50, and 70 oCF wt.%) to the olefin film and fabricated into oCFOCs by press molding for 3 min at a temperature of 180 °C with a pressure of 10 tons. Molding characteristics and aging stabilities of the oCFOCs depending on oCF contents were evaluated.

#### 2.2.2. Tensile Strength Comparison between Neat Carbon Fiber (CF) and Outdated Waste Carbon Fiber (oCF)

Single-fiber tensile tests (ASTM D3822) were performed to measure the tensile strengths of neat CF and oCF via a universal testing machine (H1KS, Lloyd, UK) operating at a 1 mm/min test speed. The diameters of neat CF and oCF were measured using a reflection microscope (JSM-7100F, Jeol Co., Ltd., Tokyo, Japan). A total of 20 tensile tests were performed to obtain and analyze statistical data via bimodal Weibull distribution function (*F* (*t*)):(1)$$F\left(t\right)=\;1-\;P,\;P=\;pexp\left[-{\left(\frac{t}{\alpha_{1}}\right)}^{\beta_{1}}\right]+qexp\left[-{\left(\frac{t}{\alpha_{2}}\right)}^{\beta_{2}}\right]\;\;p\;+\;q\;=\;1$$
where *t* is the tensile strength of fiber specimens, *P* is the probability the fiber fracture occurs at or less than a specific tensile strength, *t*, *p* and *q* are probability parameters representing low- and high-strength population portions, and *β*_1_, *α*_1_, *β*_2_, and *α*_2_ are shape and scale parameters for the low- and high-strength population portions, respectively [27].

#### 2.2.3. Evaluation of Melting Indices (MIs) of Olefin Resins and Spreading of the Resins into the oCF Tows

The MIs of olefin resins were characterized by the melt flow index measurement system (Model 6 Advanced Melt Flow System, RAY-RAN test equipment Ltd., Nuneaton, UK) based on ASTM D 1238. The amount of an individual olefin resin that was discharged for one hour under the applied weight of 2.16 kg at 210 °C was measured to evaluate the MI of each type of resin. The measurements to obtain the MI value of a resin were conducted five times, and the mean value was taken and used for relative comparisons among the resins.

In order to compare the ability of olefin resins to spread into the oCF tow according to their compositions, each type of olefin film was placed on a woven oCF fabric in the hot press and then pressurized to 3 MPa at 210 °C for 3 min. The surfaces of the manufactured oCF/olefin film specimens were observed to investigate the degree to which the resins were impregnated into the oCF fabrics with a reflection microscope (AM7013M-FIT, Dino lite Ltd., Taiwan). The results from resins with different compositions were compared to one another.

#### 2.2.4. Evaluation of Thickness Variation, Flexural Strength, Impact Strength, and Interfacial Shear Strength of oCFOCs, and Analysis of the Chemical Functional Group of Olefin Resins According to Aging Days and Olefin Resin Types

The changes in physical properties of oCFOCs were evaluated in terms of aging days (1, 2, 3, and 7 days) under specific aging conditions (120 °C, 95% humidity, and a pressure of 0.8 MPa). The visual changes in the thicknesses of oCFOCs due to the aging were assessed by preparing 20 oCFOC specimens of 20 × 20 mm^2^ and measuring their thicknesses as a function of aging days with a Vernier caliper.

The measurements of flexural strength and impact strength of oCFOCs depending on aging days were performed using a UTM (AG-250kNK, Dongil-Shimazu Ltd., Tokyo, Japan) and an Izod impact tester (IT504, Tinius Olsen Ltd., Redhill, UK), based on ASTM D 790 and ASTM D 256. Before the measurements, the specimens were kept at room temperature for 24 h.

A microdroplet pullout test was carried out to measure interfacial shear strength (IFSS). A polypropylene (PP) resin droplet was deposited on a single outdated carbon fiber (oCF). Embedded lengths were set to about 150 μm with a standard deviation of about 15 μm. The IFSS was calculated from the measured pullout force, *F*, using the following equation:(2)$$IFSS=\frac{F}{\pi D_{f}L}$$
where *D_f_* and *L* are the diameter and embedded length of the fiber in the matrix. The universal testing machine was used to pull out the single oCF with the deposited resin drop on it at a 1 mm/min test speed.

The analysis of the functional groups of olefin resins according to aging days was conducted by the ATR-FTIR (Frontier, PerkinElmer, Los Angeles, CA, USA).

## 3. Results and Discussion

### 3.1. Comparison of Tensile Strength between Neat Carbon Fiber (CF) and Outdated Waste Carbon Fiber (oCF)

As a starting point of this research on the feasibility of fabricating oCF-reinforced composites with olefin resins, the tensile strengths of single-fiber neat CF and oCF were obtained experimentally. Figure 2 shows the Weibull distribution of the tensile strengths of single-fiber neat CF and oCF. Overall, neat CF has a slightly larger value of minimum tensile strength compared to oCF, and a slightly smaller distribution range between the highest and lowest tensile strengths, suggesting that neat CF is of slightly better quality than oCF. This small degradation of tensile strength observed in oCF occurred over time. For example, when neat CF with an average tensile strength of 3300 MPa was left at room temperature for four years, which is more than the warranty period of two years, it became oCF, and the average tensile strength decreased to 3100 MPa. This change is considered to be due to the effect of the degradation of the silane layer present on the fiber surface; the degradation of the silane layer can happen due to the crosslinking between the end reaction groups, which reduces the available number of functional groups, or due to the alteration of the structure of the silanol group bonded with CF by hydrolysis and condensation under thermal and humid conditions [28]. Initially, neat CF had a smooth surface as seen in the inset of Figure 2. As it became oCF over time, the fiber surface also became rougher, dust became stuck to it, and the quality somewhat deteriorated. 

As described above, when tensile strength is simply compared between the single CF and oCF, a slight decrease of about 6% is observed on the oCF, suggesting the validity or possibility of using it rather than simply discarding it as waste.

### 3.2. Comparison of Properties of Olefin Resins Spreading into the oCF Tows with Their Melting Indices (MIs)

Figure 3 shows the MIs of the three different types of olefin resin versus the depths to which the resins spread (or impregnated) into the oCF tows. As shown, the HDPE vinyl bags exhibited the lowest MI while PP files exhibited the highest, since the MI is inversely proportional to the molecular weight (MW) of a resin. In addition to MW, the structure of HDPE, which has a higher crystallinity than PP with CH_3_ side chains, contributes to this difference in MI value. Also, as seen by red contours in the inset images, which represent the range of the resins impregnated on the backside of the oCF, the fluidity of PP was superior to that of the PE (HDPE vinyl bags and LDPE zipper bags) resins. This result suggests that PP’s spreading into the oCF tows will be superior compared to HDPE and LDPE in the fabrication of oCFOCs.

### 3.3. Molding, Flexural and Impact Properties of oCFOCs According to Aging Days under Hot and Humid Conditions

Figure 4 shows the thicknesses of as-prepared oCFOC specimens with specific oCF contents and the variation in their thickness depending on aging days. For the as-prepared oCFOCs made with HDPE, as the resin ratio to oCF increased, i.e., the oCF content decreased, the thicknesses of the specimens increased considerably due to the poor impregnation properties as seen in Figure 4a; although the specimens were produced by a mold with a thickness of 3.5 mm, the thicknesses after the molding increased to about 5.5 mm on average, indicating the HDPE resin was not suitable. However, in contrast to the as-prepared oCFOCs made with HDPE, oCFOCs made with LDPE or PP were fabricated with stable moldability, keeping the thicknesses of approximately 3.5 mm at various CF contents as seen in Figure 4b,c. The stable moldability that LDPE and PP provide allows oCFOCs to be fabricated with these resins and various oCF contents properly.

Although LDPE and PP showed stable moldability, unlike HDPE, the tendency of thickness variations in oCFOCs fabricated with these three different resins was similar, rather than different, until three aging days. As seen here, the oCFOC specimens manifesting a smaller initial thickness with increasing oCF contents exhibited decreasing thicknesses as the aging was applied until three aging days. This occurred due to the spreading of the olefin resin into the oCF tow. The results imply that the interfacial condition between the oCF and resin could be improved, as compared to the initial state, by the aging conditions of high temperature and high humidity (120 °C and 95 % humidity under a pressure of 0.8 MPa) until three aging days.

Overall, after three aging days, the thicknesses of the oCFOC specimens were increased except for the oCFOCs with PP; this is considered to have happened when these olefin resins became unstable due to overhydration in the prolonged aging conditions. As seen in Figure 4d, these thickness changes were reflected in the internal structures of the composites with oCF/HDPE, oCF/LDPE, and oCF/PP; a more detailed internal structure with oCF, resin, and a void is shown for the case of oCF/PP in Figure 4e. The extent of interlayer expansion and void formation decreases in the order of HDPE, LDPE, and PP, as shown in Figure 4d, and is highly correlated with the MI values shown in Figure 3. For oCF/HDPE or oCF/LDPE, of which the resins had lower MIs than PP, the impregnation of the resin in oCF was much poorer than that in oCF/PP. This leads to the generation of more voids internally at the interface between oCF and the resin, and more expansion of thickness externally for oCF/HDPE or oCF/LDPE than for oCF/PP.

The results suggest that the oCFOCs fabricated with HDPE and LDPE will have an issue regarding dimensional change for use in somewhat high-temperature and -humidity conditions unless the heat resistance is improved, while oCFOCs fabricated with PP can be much more resistant to dimensional change.

Next, flexural and impact strengths of oCFOC specimens according to aging days were examined. Figure 5 displays the changes in flexural strength of the specimens depending on aging days. Flexural strength increased until three aging days and then, overall, decreased for all three types of olefin resin. Specimens of oCF 45 wt.% showed the highest flexural strength on three aging days, regardless of the type of used resin. The increase in flexural strength is considered to be due to the improved spreading of the resins into the oCF tows, subsequently improving the interfacial properties of oCF and resin. Among the three types of resin, PP increased the flexural strength of oCFOCs more than 1.5 times that of HDPE or LDPE, suggesting that PP is better than the other resins for fabricating oCFOCs.

Figure 6 shows the impact strength of oCFOC specimens according to aging days. The impact strength was relatively high, as seen here, although the flexural strength was lower than that of other composites [29]. According to aging days, the variation in impact strength had a trend similar to that of flexural strength seen in Figure 5. As seen, overall, the specimens showed the highest impact strength on three aging days for the three types of resin, and specimens with 45 wt.% oCF exhibited the highest impact strength on three aging days, regardless of the resin type. Based on both flexural and impact strength results, it was confirmed that oCFOCs fabricated with PP resin are the most suitable.

### 3.4. Properties of the Chemical Functional Group of Olefin Resins According to Aging Days

To achieve insight into the changes of chemical functional group that might happen in the resins of oCFOCs during the aging, in which changes in mechanical properties occurred in the composites, attenuated total reflectance Fourier transform infrared spectroscopy (ATR-FTIR) measurements were conducted for the three types of resin according to aging days. Figure 7 shows the results of the ATR-FTIR, which provides information about the chemical functional group of the three types of resin according to aging days. As seen in Figure 7a,b, HDPE or LDPE resin shows very slight changes with aging in the vicinity of 1100–1200 cm^−1^, compared to the initial state. The peak of 1180 cm^−1^ represents the C-O functional group, but only manifests a very slight signal which does not show clearly regardless of whether the C-O group is present or not. This indicates that HDPE or LDPE still had C-H as the primary functional group.

In contrast, as seen in a dotted oval for one day in Figure 7c, a clear C-O peak was already formed in PP in just one aging day. The peak was produced by the oxidation of the functional group of PP, as shown schematically in Figure 7d. This suggests that PP is much more susceptible to oxidation than HDPE or LDPE under the aging conditions of high temperature and high humidity (120 °C and 95% humidity under a pressure of 0.8 MPa) [30]. The C-O peak further developed and showed the highest point in three aging days. The developed C-O groups can form hydrogen bonds between the parts of PP with these functional groups and the existing silane layer of oCF as schematically shown in Figure 8, subsequently improving the mechanical properties of oCFOCs demonstrated in Figure 5 and Figure 6. Then, the peak was observed to disappear in seven aging days. C-O bonds rather recovered to the original PP structure (Figure 7d) or PE structure at this longer aging time. This result might suggest that the formed hydrogen bonds between the resin and oCF would be lost, resulting in the production of the degraded mechanical properties of the oCFOCs in seven aging days.

### 3.5. Interfacial Properties of oCFOCs According to Aging Days

Figure 9 expresses the interfacial properties of as-prepared and aged oCFOCs based on the data of oCFOC thickness and the results of ATR-FTIR measurements of the olefin resins schematically. When oCFOCs are produced just by pressurizing an olefin resin and oCF tow, it is difficult to form a strongly cohesive interface due to the dust (red circles in Figure 1a) and unavailability of the functional group of the resin reacting with the oCF surface (Figure 9a). The aging conditions, having heating and humidity together with low pressure, could impregnate the resin into the oCF tow effectively due to the increased softness of the resin due to the aging. This subsequently causes a decrease in the thickness of the oCFOC and increases the cohesive force between the resin and oCF (Figure 9b,c). Because the resin does not have functional groups such as OH, the cohesive force derives from the mechanical interlocking between the resin and oCF. However, unlike oCFOCs with HDPE or LDPE, for the oCFOC with PP, the cohesive force can be assisted by the hydrogen bond between the resin and oCF, as schematically expressed in Figure 8, drawn on the basis of the results in Figure 7. Adhesion between the resin and oCF tow became optimal in three aging days for the oCFOCs of most oCF contents, regardless of the resin type (Figure 9c). However, as the aging proceeded further, the resin was excessively hydrated. This caused a decrease in the cohesive strength between the resin and oCF followed by the increase of oCFOC thickness, and subsequently, the stability of oCFOCs was lowered (Figure 9d).

Figure 10 shows interfacial shear strength (IFSS) of oCF/PP composites with different PP conditions measured by the microdroplet pullout test. As shown by the black line in Figure 10, the IFSS of the oCF/PP composite with neat PP was 8.5 MPa (see Section 2.2.4 of 2. Materials and Methods for the IFSS formula). However, the IFSS of the oCF/PP composite with PP with three aging days (red line) was 14.2 MPa, a value improved by 67.1%. This enhancement is attributed to the improved impregnation of PP under the aging condition and could be due to hydrogen bonds formed between the parts of PP with C-O groups and the existing silane layer of oCF. The results suggest that incubating oCF/PP composites in specific high-temperature and -humidity conditions could be a novel method to improve oCF/PP interfacial properties.

## 4. Conclusions

In this work, outdated-waste-carbon-fiber-reinforced olefin composites (oCFOCs) were fabricated using commercial outdated waste carbon fiber (oCF) and three kinds of easily disposable polyolefin resins (high-density polyethylene (HDPE) vinyl bags, low-density polyethylene (LDPE) zipper bags, and polypropylene (PP) files). To find the feasibility of the fabricated oCFOCs as durable composites, several properties of the oCFOCs were evaluated. Among the respective resins, PP exhibited the most excellent properties of impregnation into the oCF tow, commensurate with it scoring the highest melting index (MI) value; these properties also led to the fabrication of oCFOCs with better moldability, containing fewer voids in the internal structure, and the improved cohesion between the resin and oCF, compared to HDPE or LDPE. Under the aging treatment (120 °C, 95% humidity, and 0.8 MPa), changes in thickness and mechanical properties (flexural and impact strength) were manifested according to the aging days, due to the permeation of the resin into the oCF tow and hydration. The oCFOC with 45 wt.% oCF content and three aging days exhibited the highest flexural and impact strength, regardless of the type of resin. Among the oCFOCs, the one fabricated with PP exhibited the highest flexural and impact strength, maintaining good dimensional stability due to PP having much better impregnation properties than the other resins. The excellent impregnation properties of PP were also reflected in the interfacial shear strength (IFSS) of the oCF/PP composite by the microdroplet pullout test. The IFSS of the composite with PP aged for three days was about 70% higher than that of the composite with neat PP; 14.2 MPa (PP aged for three days) vs. 8.5 MPa (neat PP). The enhancement of the IFSS is attributed to the improved impregnation of PP and could be due to hydrogen bonds between the parts of PP with C-O groups and the existing silane layer of oCF due to the aging treatment.

Based on the results demonstrated above, it is suggested that PP is the most suitable resin for the fabrication of durable oCFOCs, and that oCFOCs with good mechanical properties can be achieved with PP and specific aging treatments under the conditions of high temperature and humidity. Thus, we propose that if the olefin-resin-based carbon fiber composite products are sorted out quickly through proper recycling process, similar to the way the ‘Precious plastic project’ works, the manufacturing of PP-based composite products using recycled carbon fiber could be a desirable carbon fiber waste management route.

## Figures and Tables

**Figure 1 polymers-13-02938-f001:**
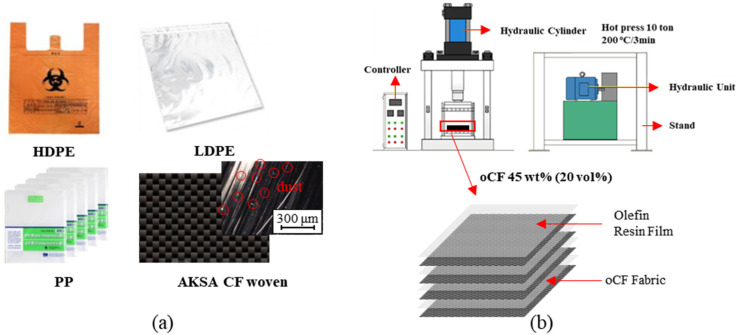
Experimental materials of (**a**) film-type olefin resins (HDPE, LDPE, and PP), outdated carbon fiber (oCF), and (**b**) schematics on the fabrication of outdated-waste-carbon-fiber-reinforced olefin composites (oCFOCs) by the compressive molding machine.

**Figure 2 polymers-13-02938-f002:**
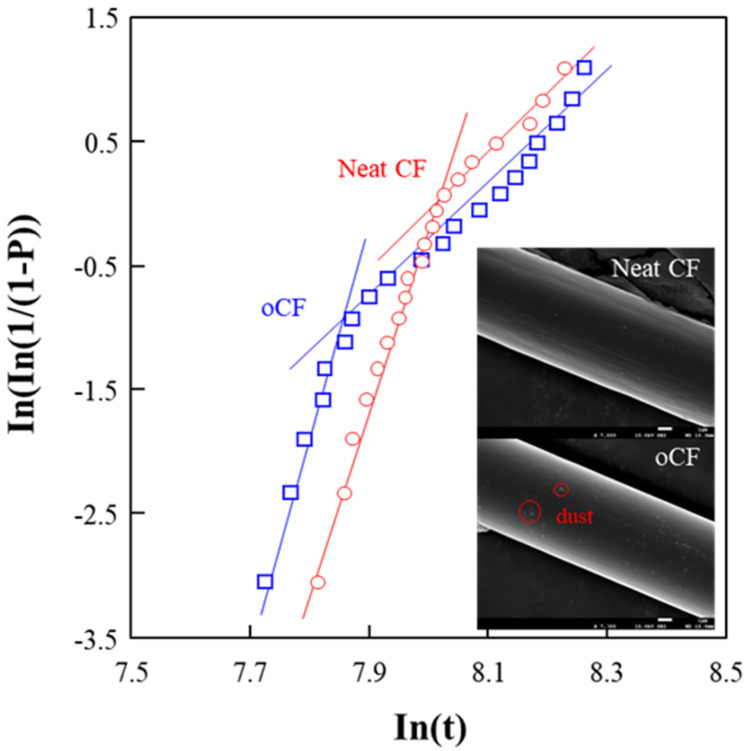
Weibull distribution of the tensile strengths of neat CF and oCF. t is the tensile strength and P is the probability the fiber fracture occurs at or less than a specific tensile strength.

**Figure 3 polymers-13-02938-f003:**
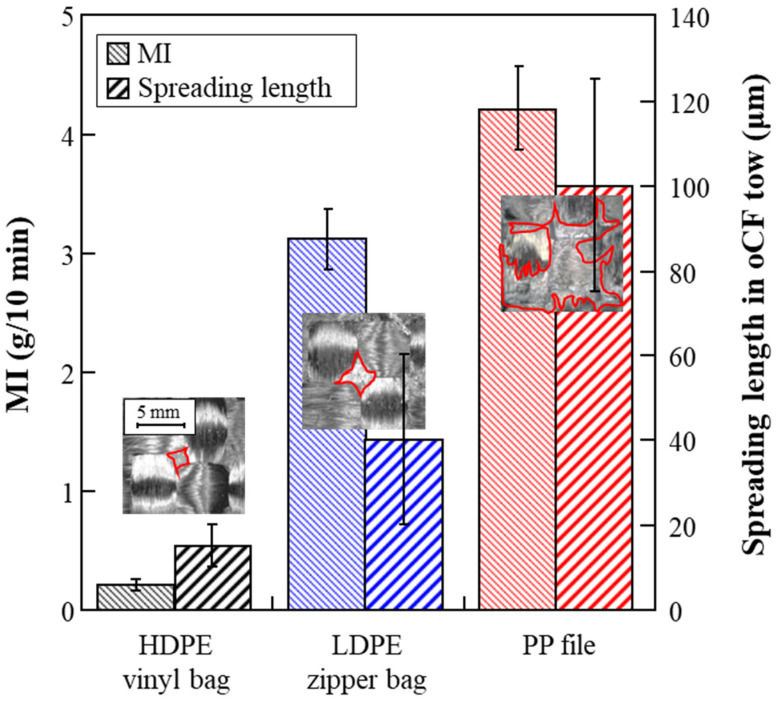
Melting indices (MIs) and spreading properties of three olefin resins: HDPE, LDPE, and PP.

**Figure 4 polymers-13-02938-f004:**
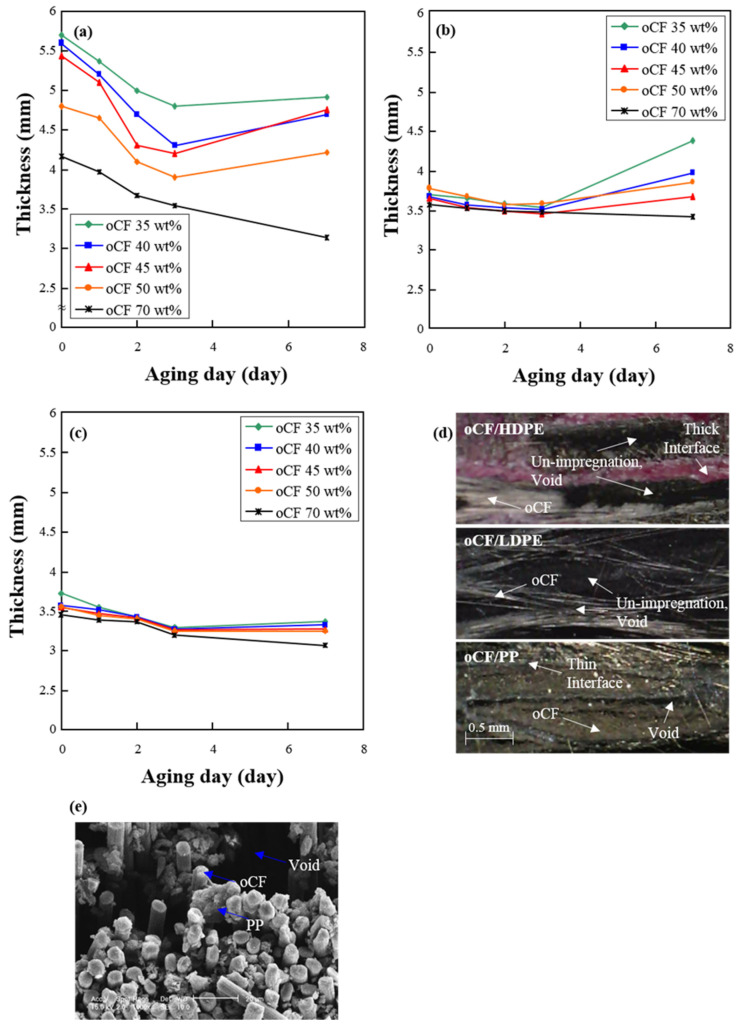
Variation in the thicknesses of oCFOCs by aging days with (**a**) HDPE, (**b**) LDPE, and (**c**) PP, and (**d**,**e**) cross-sectional images of the 45 wt.% oCFOC with seven aging days.

**Figure 5 polymers-13-02938-f005:**
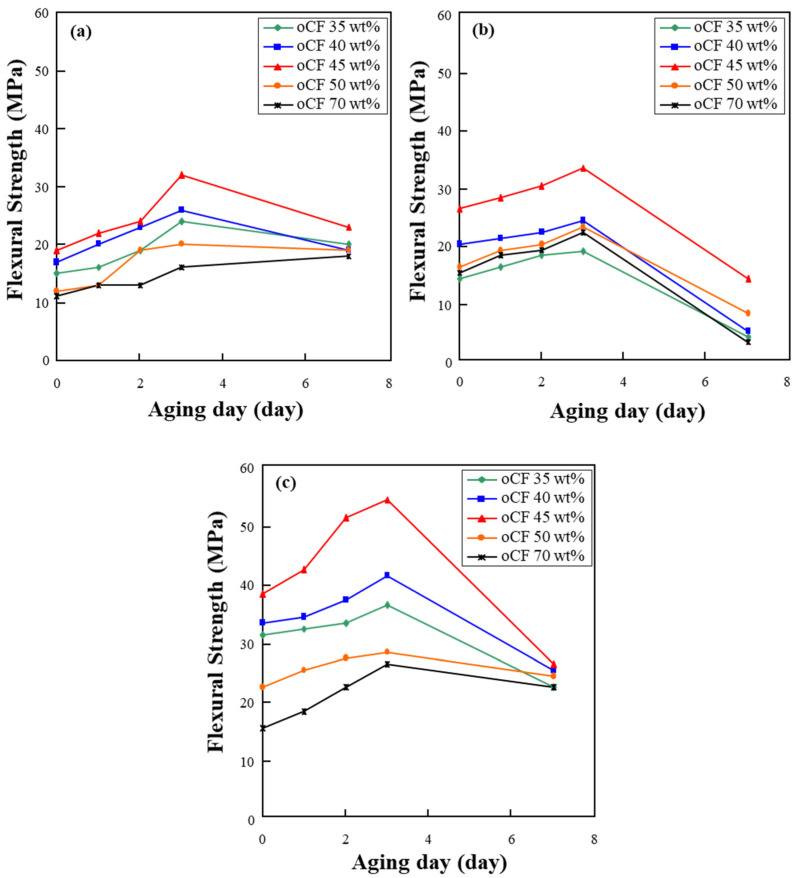
Variation in flexural strength of oCFOCs by aging days: (**a**) HDPE, (**b**) LDPE, and (**c**) PP.

**Figure 6 polymers-13-02938-f006:**
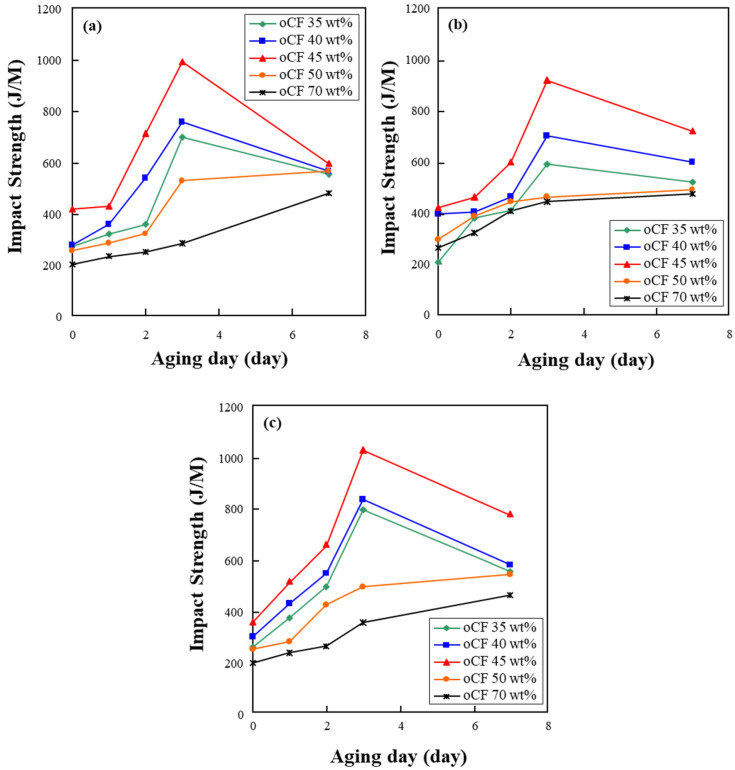
Variation in impact strength of oCFOCs by aging days: (**a**) HDPE, (**b**) LDPE, and (**c**) PP.

**Figure 7 polymers-13-02938-f007:**
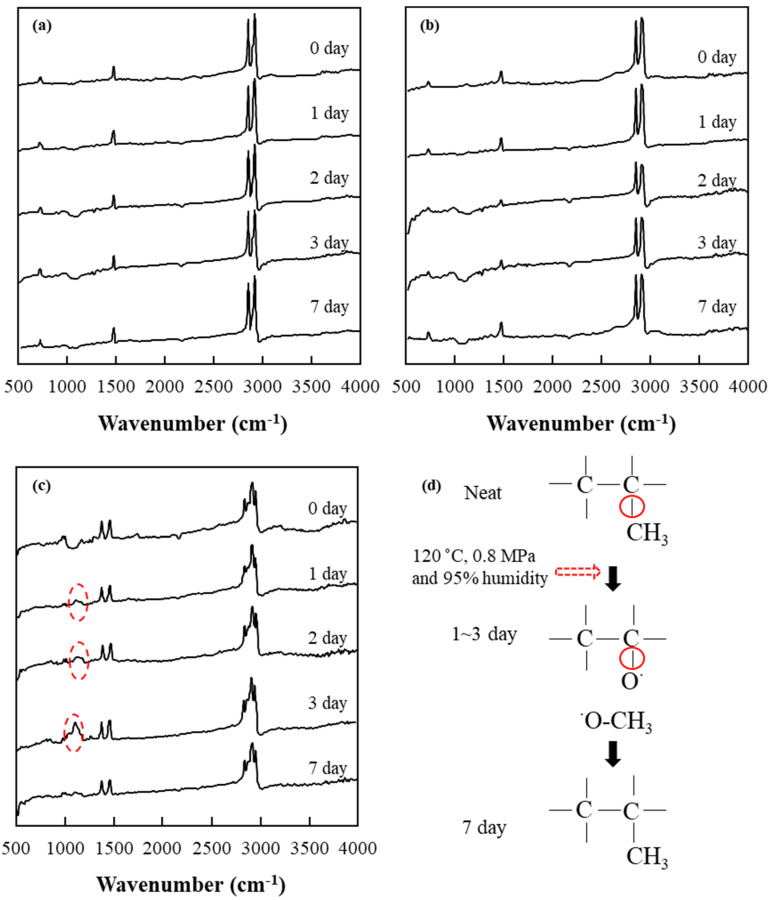
ATR-FTIR spectra of (**a**) HDPE, (**b**) LDPE, and (**c**) PP, and (**d**) a schematic expression of chemical reactions in PP according to aging days.

**Figure 8 polymers-13-02938-f008:**
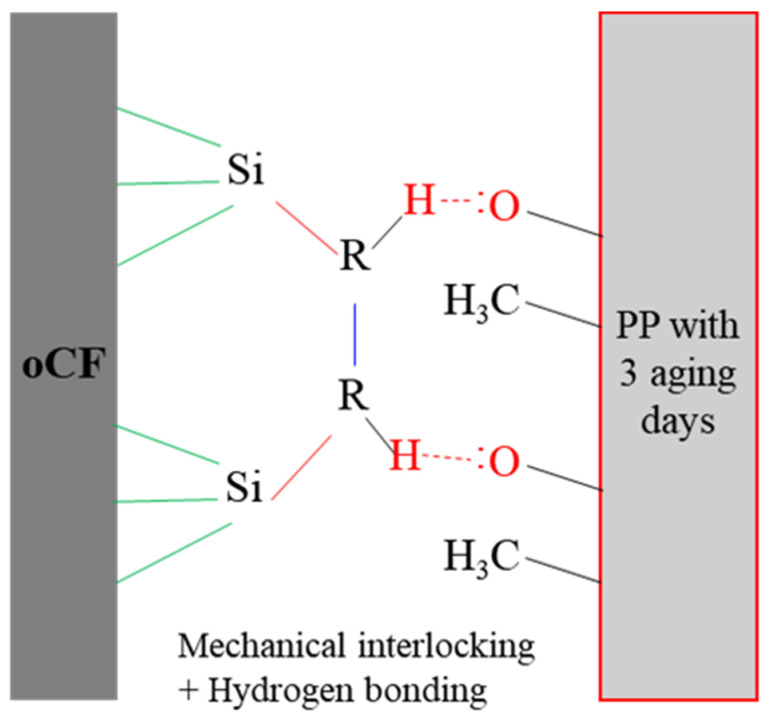
Schematic of the hydrogen bonds between oxidized PP and silane layer of oCF.

**Figure 9 polymers-13-02938-f009:**
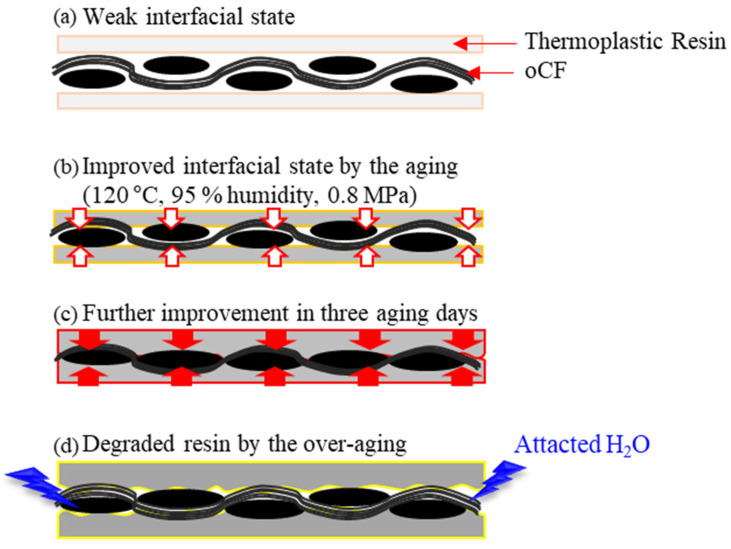
Schematic showing the interfacial properties of (**a**) as-prepared and (**b**–**d**) aged oCFOCs according to aging days.

**Figure 10 polymers-13-02938-f010:**
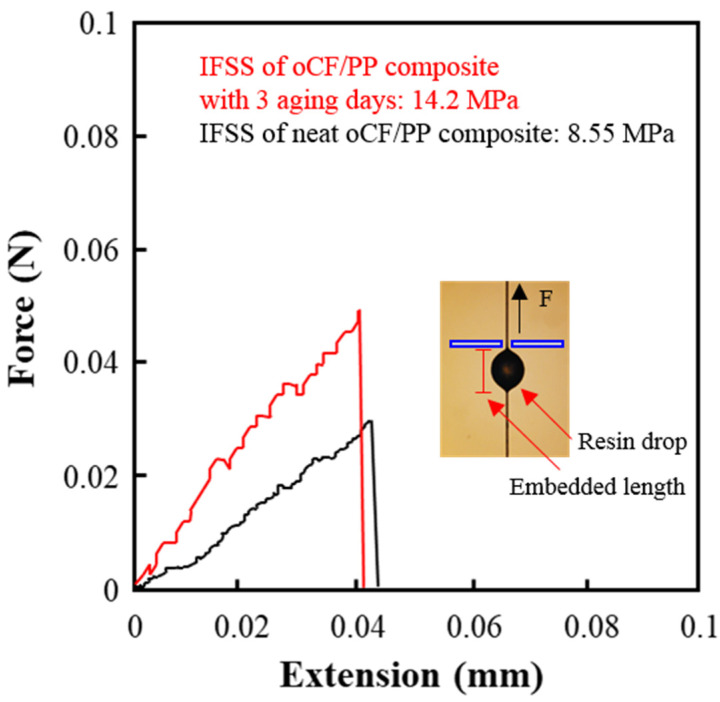
Results of the microdroplet pullout test of oCF/PP composites with PP aged for three days or with neat PP.

## Data Availability

The data presented in this study are available on request from the corresponding author.

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
