# Peer review of "An Investigation on the Feasibility of Fabricating Composites Using Outdated Waste Carbon Fibers and Easily Disposable Polyolefin Resins"

_polymers, 2021, doi:10.3390/polym13172938_

Round 1
Reviewer 1 Report
Authors reported utilizing waste carbon fiber as filler to reinforce polyolefin materials. Authors screened three different polymers (PP, HDPE, and LDPE) and various percentage. Thus, composite materials were studied by different properties such as tensile strength, melting index, moldability, thickness change, flexural and impact strength. In addition, authors studied aging effect on these composite materials. Lastly, authors investigated reason whey PP worked better than PE upon aging, using IR spectroscopy and microdroplet pullout test. This manuscript has clear experiment design and is suitable for the scope of Polymers. I would like to recommend for acceptance with major revision.
- Why interwoven CF was investigated? Could you also try chopped CF?
- In the context of sustainability and cyclic economics, in order to prove feasibility, could you also try oCF from other vendors?
- What kind of silane is on the surface of CF? What’s the mechanism for this silane degradation?
- What is the hydrogen bonding between oxidized PP and silane? Could you please draw a scheme? Do you have any reference?
Author Response
Authors’ responses: below, we respond to the reviewer #1’ each comment by point-by-point.
Reviewer #1’s full Comments to authors:
Authors reported utilizing waste carbon fiber as filler to reinforce polyolefin materials. Authors screened three different polymers (PP, HDPE, and LDPE) and various percentage. Thus, composite materials were studied by different properties such as tensile strength, melting index, moldability, thickness change, flexural and impact strength. In addition, authors studied aging effect on these composite materials. Lastly, authors investigated reason whey PP worked better than PE upon aging, using IR spectroscopy and microdroplet pullout test. This manuscript has clear experiment design and is suitable for the scope of Polymers. I would like to recommend for acceptance with major revision.
- Why interwoven CF was investigated? Could you also try chopped CF?
- In the context of sustainability and cyclic economics, in order to prove feasibility, could you also try oCF from other vendors?
- What kind of silane is on the surface of CF? What’s the mechanism for this silane degradation?
- What is the hydrogen bonding between oxidized PP and silane? Could you please draw a scheme? Do you have any reference?
(General comment of the reviewer)
Authors reported utilizing waste carbon fiber as filler to reinforce polyolefin materials. Authors screened three different polymers (PP, HDPE, and LDPE) and various percentage. Thus, composite materials were studied by different properties such as tensile strength, melting index, moldability, thickness change, flexural and impact strength. In addition, authors studied aging effect on these composite materials. Lastly, authors investigated reason whey PP worked better than PE upon aging, using IR spectroscopy and microdroplet pullout test. This manuscript has clear experiment design and is suitable for the scope of Polymers. I would like to recommend for acceptance with major revision.
Our Responses: We greatly appreciate the reviewer’ good evaluation and constructive comments and questions. We have carefully considered all of these comments and questions and dealt with them in our responses to the comments of the reviewer and updated them in the revised manuscript.
- Why interwoven CF was investigated? Could you also try chopped CF?
Our Responses: As we described, the purpose of this study was to investigate the feasibility of fabricating carbon fiber reinforced olefin composites using outdated waste, i.e. discarded CF (oCF) and easily-disposed of polyolefin resins as a way of oCF recycling. When we conducted this preliminary study for fabricating outdated waste carbon fiber reinforced olefin composites (oCFOC), the oCF we only had was a fabric-type, interwoven CF and the oCFOC was manufactured using the sheet molding process, which was the easiest way to manufacture composites. We did not try chopped CF at that time. However, currently, we are collecting waste for further research on chopped oCF or chopped rCF. As commented by the reviewer, we consider conducting research on this topic using chopped CF will be an important work for this type of research. We appreciate the reviewer for this good suggestion.
- In the context of sustainability and cyclic economics, in order to prove feasibility, could you also try oCF from other vendors?
Our Responses: This study was conducted using waste carbon fibers we had in our laboratory. We agree with the reviewer’s comment. To assess feasibility in more general aspects, we think we have to obtain results using oCF from different manufacturers for further study. We think this will provide information about commons and differences in properties such as mechanical properties and formability depending on the used oCF from different sources, rendering assessment of feasibility in a more general view.
- What kind of silane is on the surface of CF? What’s the mechanism for this silane degradation?
Our Responses: The carbon fiber (model number: A-42) manufactured by DOWAKSA was used as described in the MATERIALS AND METHODS section. In most cases, information about the chemical structure of silane that is on the surface of CF in such a tiny amounts is not informed by the manufacturer. However, usually, the silane has either NH2 or COC epoxy groups at its terminal. Below is shown the most common silane structure.
<The most common silane structre>
Thus, here we describe the generally expected silane degradation mechanism based on the result from references (Celina et al., characterization and degradation studies of peroxide and silane crosslinked polyethylene. Polymer Degradation and Stability 48 (1995) 297-312) rather than specifically and exactly designated mechanism confined to our case because the information about the silane chemical structure on the used oCF is not available as we described above.
Silane has a functional group, i.e., end reaction group R as shown in the above or below Figure.
When it is exposed to a thermal environment containing water, the saline crosslinking as schematically seen in the below Figure can happen and the available number of R is reduced, losing some of its function (Ahmed et al., Polymer, 11, 537, 2019).
Also, the hydrolysis and condensation can alter the structure of the silanol group bonded with CF, as seen below, deteriorating the function of silane.
Following the reviewer’s question, we have updated a part originally written as
“This change is considered to be due to the effect of degradation of the silane layer present on the fiber surface.” at page 5 (from 6th line from the top) in the manuscript as follows with ref. (Ahmed et al., Polymer, 11, 537, 2019)):
“This change is considered to be due to the effect of degradation of the silane layer present on the fiber surface; the degradation of the silane layer can happen by the crosslinking between the end reaction groups which reduces the available number of the functional groups, or alteration of the structure of silanol group bonded with CF by the hydrolysis and condensation under a thermal and humid condition as seen below [28].
- What is the hydrogen bonding between oxidized PP and silane? Could you please draw a scheme? Do you have any reference?
Our Responses: Although PP is stable and does not have a functional group such as OH, it can form unstable radicals when exposed to high temperature and humidity (F. H. Winslow, Pure Appi. Chem., Vol. 49, pp. 495 - 502 (1977); this ref. is now added to the revised manuscript as ref.30). The FT-IR results (Fig. 7) in this study confirmed the results.
The oxidized PP can make hydrogen bond with silane. We express this using the schematic below.
We have now updated the manuscript by placing this scheme as Fig. 8 in the relevant place in the manuscript. At page 10, from second line from the bottom, originally it was written as “The developed C-O groups can form hydrogen bonds between the parts of PP with these functional groups and the existing silane layer of oCF, subsequently improving the mechanical properties of the oCFOC demonstrated in Figures 5 and 6.
Now in this revised version, it is written as “The developed C-O groups can form hydrogen bonds between the parts of PP with these functional groups and the existing silane layer of oCF as schematically shown in Fig. 8, subsequently improving the mechanical properties of the oCFOC demonstrated in Figures 5 and 6.
And we have now added the reference (F. H. Winslow, Pure Appi. Chem., Vol. 49, pp. 495 - 502 (1977)) to the part “This suggests that PP is much more susceptible to oxidation than HDPE or LDPE un-der the aging condition of high temperature and high humidity (120 °C and 95 % hu-midity under a pressure of 0.8 MPa).” originally written at page 10 (second line from the bottom).
And also another relevant place has been updated.
Originally, from 7th line from the top of page 12, it was written “The aging conditions having heating, humidity, together with low pressure could impregnate the resin into the oCF tow, subsequently causing a decrease in the thickness of the oCFOC and increasing cohesive force between the resin and oCF (Figs. 8(b) and 8(c)). For the oCFOC with PP, the cohesive force can be assisted by the hydrogen bond between the resin and oCF.”
Now in this revised version, it is written as “The aging conditions having heating and humidity together with low pressure could impregnate the resin into the oCF tow effectively due to the increased softness of the resin by the aging. This subsequently causes a decrease in the thickness of the oCFOC and increases the cohesive force between the resin and oCF (Figs. 9(b) and 9(c)). Because the resin does not have functional groups such as OH, the cohesive force derives from the mechanical interlocking force between the resin and oCF. However, unlike oCFOC with HDPE or LDPE, for the oCFOC with PP, the cohesive force can be assisted by the hydrogen bond between the resin and oCF, as schematically expressed in Fig. 8 drawn on a basis of the results in Fig. 7.”

Reviewer 2 Report
Manuscript entitled “An investigation on the feasibility of fabricating composites using outdated waste carbon fibers and easily-disposed of polyolefin resins” submitted by Dong-Jun Kwon, Kang Rae Cho and Hyoung-Seock Seo, can be accepted for publication in Polymers Journal, after a major revision.
Here is a list of my specific comments:
- General comment 1: The utility of this study should be clearly highlighted in the manuscript.
- General comment 2: Pay attention on the interpretation of the experimental results. A detailed interpretation of these increases the importance of this study.
- Page 5: “For example, when neat CF…”. These results were obtained experimental??? If yes, then this aspect should be mentioned, as well as the time period required for such degradation. If no, a reference should be added.
- Page 5: “Interfacial shear strength (IFSS) evaluation…”. Delete this paragraph.
- Page 5, 3.2. Comparison of properties of olefin resins spreading into oCF tows with their melting index (MI): The results included in this section should be more detailed discussed.
- Page 6: “As seen in Fig. 4(d), these thickness changes…”. These observations should be detailed.
- Page 12, 3.5. Interfacial properties of oCFOC according to aging days: The experimental results included in this section should be better explained.
- Page 13, 4. CONCLUSION: This section is too long and should be shortened. Delete (1), (2),… and provide a properly description of the most important experimental results and findings included in this study.
Author Response
Authors’ responses: below, we respond to the reviewer #2’ each comment by point-by-point.
Reviewer #2’s full Comments to authors:
Manuscript entitled “An investigation on the feasibility of fabricating composites using outdated waste carbon fibers and easily-disposed of polyolefin resins” submitted by Dong-Jun Kwon, Kang Rae Cho and Hyoung-Seock Seo, can be accepted for publication in Polymers Journal, after a major revision.
Here is a list of my specific comments:
- General comment 1: The utility of this study should be clearly highlighted in the manuscript.
- General comment 2: Pay attention on the interpretation of the experimental results. A detailed interpretation of these increases the importance of this study.
- Page 5: “For example, when neat CF…”. These results were obtained experimental??? If yes, then this aspect should be mentioned, as well as the time period required for such degradation. If no, a reference should be added.
- Page 5: “Interfacial shear strength (IFSS) evaluation…”. Delete this paragraph.
- Page 5, 3.2. Comparison of properties of olefin resins spreading into oCF tows with their melting index (MI): The results included in this section should be more detailed discussed.
- Page 6: “As seen in Fig. 4(d), these thickness changes…”. These observations should be detailed.
- Page 12, 3.5. Interfacial properties of oCFOC according to aging days: The experimental results included in this section should be better explained.
- Page 13, 4. CONCLUSION: This section is too long and should be shortened. Delete (1), (2),… and provide a properly description of the most important experimental results and findings included in this study.
(General comment of the reviewer)
Manuscript entitled “An investigation on the feasibility of fabricating composites using outdated waste carbon fibers and easily-disposed of polyolefin resins” submitted by Dong-Jun Kwon, Kang Rae Cho and Hyoung-Seock Seo, can be accepted for publication in Polymers Journal, after a major revision.
Our Responses: We greatly appreciate the reviewer’ constructive comments and suggestions shown below. We have carefully considered all of them and dealt with them in our responses to the comments of the reviewer and updated them in the revised manuscript.
- General comment 1: The utility of this study should be clearly highlighted in the manuscript.
Our Responses: As the reviewer recognizes, although it was not a very detail, we described the purpose or the utility of this study in the Introduction and Conclusion parts: At page 2, starting from 11th line from the bottom of the Introduction part of the original manuscript, it writes “Thus, the results show that as a good way of oCF recycling, durable oCFOC with good mechanical properties could be manufactured by choosing a proper matrix (such as PP for this study) together with optimal oCF contents under suitable aging conditions.
At page 14, starting from 15th line from the top of the Conclusion part of the orinal manuscript (i.e., concluding remark), it writes “Thus, we propose that if the olefin resin-based carbon fiber composite products are sort out quickly through proper recycling process like the way ‘Precious plastic project’ works, the manufacture of PP-based composite products using recycled carbon fiber could be one of the desirable carbon fiber waste management routes.”
Following the reviewer’s comment, we have modified the description (“Thus, the results show….. under suitable aging conditions.”) written in that Introduction to make it clearer in this revision. (We consider the concluding remark is fine as it is, so we did not update more). Now it is modified as “Thus, the results show that durable oCFOC with good mechanical properties could be manufactured by choosing a proper matrix (such as PP for this study) together with optimal oCF contents under suitable aging conditions. We propose that our results will serve as important preliminary data to demonstrate and support the potential of fabricating durable oCF composites, as a good way of oCF recycling.”
- General comment 2: Pay attention on the interpretation of the experimental results. A detailed interpretation of these increases the importance of this study.
Our Responses: We consider that more detailed interpretation of the experimental results is now updated in this revision, following the reviewer’s comments from Q3, Q4, Q5-Q7.
- Page 5: “For example, when neat CF…”. These results were obtained experimental??? If yes, then this aspect should be mentioned, as well as the time period required for such degradation. If no, a reference should be added.
Our Responses: Both results of neat CF and oCF were obtained experimentally. In the subsection (2.2.2.) of 2.2 Methodologies (of original manuscript at page 3), from beginning, it is described as ”Single-fiber tensile tests (ASTM D3822) were performed to measure the tensile strength of neat CF and oCF by a universal testing machine (H1KS, Lloyd, UK) operating at a 1 mm/min test speed”.
Taking the reviewer’s comment, we have now specified that these results came from the experiments by adding a word “experimentally” to the beginning sentence of subsection 3.1 of 3. RESULTS AND DISCUSSION at the bottom of page 4. It is now written as “As a starting point of this research on the feasibility of fabricating oCF reinforced composites with olefin resins, the tensile strength of single fiber neat CF and oCF was obtained experimentally.”
Apart from this, we did a mistake by describing that oCF was left at room temperature for two years there at page 5 (5th line from the top of the original manuscript). The correct sentence is that oCF was left at room temperature for four years (the warrenty period is two years). We are very sorry about failing in correct description.
Now the sentence ““For example, when neat CF…” is updated as “For example, when neat CF with an average tensile strength of 3300 MPa was left at room temperature for four years which are more than the warranty period of two years, it became oCF, and the average tensile strength was decreased to 3100 MPa.”
And the reviewer asked about the time period required for such degradation of 200 MPa of the tensile strength. Unfortunately, we did not measure the property for every some interval, rather, measured the property of carbon fiber with 4 years spent after the time of purchase and neat carbon fiber. A better understanding would have been obtained if the physical property had been evaluated every year. Please understand this point that we cannot accurately evaluate the starting time period required for such degradation of 200 MPa of the tensile strength because of this reason.
- Page 5: “Interfacial shear strength (IFSS) evaluation…”. Delete this paragraph.
Our Responses: We have deleted “Interfacial shear strength (IFSS) evaluation of oCF/PP and CF/PP composites, which will be shown in a subsection (3.5. Interfacial properties of oCFOC according to aging days) of RESULTS AND DISCUSSION further supports the validity of using it for the fabrication of low-quality oCF/olefin thermoplastic composites.” now in this revised manuscript.
- Page 5, 3.2. Comparison of properties of olefin resins spreading into oCF tows with their melting index (MI): The results included in this section should be more detailed discussed.
Our Responses: We have improved the content. Originally, from 8th line from the bottom at page 5, it is described as “Fig. 3 shows the MI of the three different types of olefin resins versus the spread (or impregnated) depths of the resins into the oCF tows. As shown, the HDPE vinyl bag exhibited the lowest MI while PP files the highest because the MI is inversely proportional to the molecular weight of a resin.”.
Now in this revision, it is written as “Fig. 3 shows the MI of the three different types of olefin resins versus the spread (or impregnated) depths of the resins into the oCF tows. As shown, the HDPE vinyl bag exhibited the lowest MI while PP files the highest because the MI is inversely proportional to the molecular weight (MW) of a resin. In addition to MW, the structure of HDPE which has higher crystallinity than PP with CH3 side-chains contributes to this difference in MI value.”
- Page 6: “As seen in Fig. 4(d), these thickness changes…”. These observations should be detailed.
Our Responses: We have modified that part following this suggestion. Originally, it was written as:
“As seen in Fig. 4(d), these thickness changes were reflected in the internal structures of the composites with oCF/HDPE, oCF/LDPE and oCF/PP; a more detailed internal structure having oCF, resin, and void is shown for the case of oCF/PP in Fig. 4(e). The extent of interlayer expansion and void formation is decreasing in the order of HDPE, LDPE, and PP. In particular, oCFOC fabricated with PP showed much improved impregnation of the resin into the fiber tow compared with that of HDPE or LDPE.
The results suggest that the oCFOC fabricated with HDPE and LDPE will have an issue regarding dimensional change for use in the condition of somewhat high temper-ature and humidity unless the heat resistance is improved while oCFOC fabricated with PP can be much more resistant to the dimensional change.”
Now we have updated it as:
“As seen in Fig. 4(d), these thickness changes were reflected in the internal structures of the composites with oCF/HDPE, oCF/LDPE and oCF/PP; a more detailed internal structure having oCF, resin, and void is shown for the case of oCF/PP in Fig. 4(e). The extent of interlayer expansion and void formation is decreasing in the order of HDPE, LDPE, and PP, as shown in Fig. 4(d), and is highly correlated with the MI values in Fig. 3. For oCF/HDPE or oCF/LDPE of which resin had a lower MI than PP, the impregnation of the resin in oCF was much poorer than that in oCF/PP. This leads to generating more voids internally at the interface between oCF and resin, and more expansion of thickness externally for oCF/HDPE or oCF/LDPE than oCF/PP.
The results suggest that the oCFOC fabricated with HDPE and LDPE will have an issue regarding dimensional change for use in the condition of somewhat high temper-ature and humidity unless the heat resistance is improved while oCFOC fabricated with PP can be much more resistant to the dimensional change.”
- Page 12, 3.5. Interfacial properties of oCFOC according to aging days: The experimental results included in this section should be better explained.
Our Responses: We have modified the relevant part following this suggestion. The modified part was originally written as:
“The aging conditions having heating, humidity, together with low pressure could im-pregnate the resin into the oCF tow, subsequently causing a decrease in the thickness of the oCFOC and increasing cohesive force between the resin and oCF (Figs. 8(b) and 8(c)). For the oCFOC with PP, the cohesive force can be assisted by the hydrogen bond between the resin and oCF. Adhesion between the resin and oCF tow was optimal in three aging days (Fig. 8(c)).” at page 12, from 7th line.
Now we have updated it as:
“The aging conditions having heating and humidity together with low pressure could impregnate the resin into the oCF tow effectively due to the increased softness of the resin by the aging. This subsequently causes a decrease in the thickness of the oCFOC and increases the cohesive force between the resin and oCF (Figs. 9(b) and 9(c)). Because the resin does not have functional groups such as OH, the cohesive force derives from the mechanical interlocking between the resin and oCF. However, unlike oCFOC with HDPE or LDPE, for the oCFOC with PP, the cohesive force can be assisted by the hydrogen bond between the resin and oCF, as schematically expressed in Fig. 8 drawn on a basis of the results in Fig. 7. Adhesion between the resin and oCF tow became optimal in three aging days for the oCFOC of most oCF contents regardless of the resin types (Fig. 9(c)).”
- Page 13, 4. CONCLUSION: This section is too long and should be shortened. Delete (1), (2),… and provide a properly description of the most important experimental results and findings included in this study.
Our Responses: We have shortened the section and now expressed it without numbering of (1), (2) in the revised manuscript.

Round 2
Reviewer 1 Report
The revision is good. I think it is good for acceptance for publication.
Reviewer 2 Report
All my previous remarks and comments have been considered into new version of the manuscript. It means that reviewed manuscript meets the criteria and in my opinion can be published as original paper in Polymers Journal.